# Understanding Parental Adherence to Early Childhood Domestic Injury Prevention: A Cross-Cultural Test of the Integrated Behavior–Change Model

**DOI:** 10.3390/bs14080701

**Published:** 2024-08-12

**Authors:** Roni M. Y. Chiu, Derwin K. C. Chan

**Affiliations:** 1Department of Social and Behavioural Sciences, City University of Hong Kong, Hong Kong, China; manychiu@cityu.edu.hk; 2Department of Early Childhood Education, The Education University of Hong Kong, Hong Kong, China

**Keywords:** self-determination theory, theory of planned behavior, home safety, multigroup structural equation modeling, multigroup confirmatory factor analysis

## Abstract

Unintentional injuries pose a significant risk to children in early years globally. In particular, toddlers and preschoolers are vulnerable to injuries that occur at home. Despite the availability of preventive measures that can greatly reduce the risks of domestic injuries, some caregivers (e.g., parents) of children in early childhood may not fully implement these safety measures due to poor behavioral adherence or low awareness of the risk of domestic injury. Therefore, it is crucial to understand how caregivers in different cultural contexts approach injury prevention in the home environment. In this multi-cultural study, we investigated the motivational and belief processes underlying childhood domestic injury prevention among a total of 2059 primary caregivers (parents/guardians) of infant and toddlers (aged 2 to 6 years) across four societies, Australia (AU; N = 500), the United States (US; N = 500), Singapore (SG; N = 507), and Hong Kong (HK; N = 552), by applying the integrated model of self-determination theory (SDT) and theory of planned behavior (TPB). Our results support the key tenets of the integrated model and demonstrated cultural invariance model pathways across the four societies studied. In particular, the positive relationships among psychological need support, autonomous motivation, socio-cognitive beliefs, intention, and behavior adherence remained constant across societies. With a multi-cultural sample, this study provides valuable insights into the similarities and differences in motivation and beliefs surrounding childhood domestic injury prevention across these four societies.

## 1. Introduction

### 1.1. Background

Unintentional injury is one of the major causes of childhood death and disability. Globally, unintentional injury accounts for 32.9% of childhood fatalities [1], and numerous others require hospitalization or must seek treatment at emergency departments [2,3,4]. It is noteworthy that a majority of these incidents occur within the home environment, particularly among younger children during the stage of early childhood [5,6,7]. Even minor domestic injuries should not be overlooked, as in almost identical circumstances such injuries could easily escalate to more severe outcomes [8]. Additionally, mild injuries can lead to adverse outcomes, such as missed school/daycare/workdays for parents, as well as disruptions of daily activities [9] and psychological distress [10]. Beyond the immediate health risks, the financial cost incurred due to medical and life-long disability is substantial [2,3,4], illustrating that childhood domestic injury is a pervasive public health concern.

Domestic injuries can be effectively minimized through the consistent adoption of safety precautions. Domestic injury prevention includes active supervision, environmental modification, and safety education for children [11]. Active supervision encompasses three dimensions (i.e., attention, proximity, and continuity) [11]. Caregivers should constantly attend to and be within close proximity of their child. Environmental modification includes limiting access to or removing hazards, such as keeping medicine cabinets out of reach [11]. Safety education refers to establishing clear behavioral rules for children to follow, like not touching knives [11]. Research suggests that a significant number of childhood deaths (estimated between 33% and 65%) and hospital admissions (over 85%) resulting from unintentional injuries could be prevented [2,12,13]. However, concerning statistics reveal that approximately one-third of caregivers failed to implement fall prevention measures (such as preventing their children from climbing or the use of belts/highchairs and safety gates) or adopt general domestic safety precautions for young children (such as the use of corner protectors, safety gates, and door knob covers) in Canada and HK, respectively [14,15]. Multiple studies have identified numerous barriers that hinder the proper implementation of childhood domestic injury prevention. These barriers encompass factors such as insufficient knowledge [16], underestimation of injury risks [17], a false perception that domestic accidents are unavoidable, and a tendency to prioritize convenience over injury risks [18,19]. Although educational interventions and home visit programs have demonstrated some success in improving caregivers’ knowledge and awareness of domestic injuries, their long-term impact on injury prevention has been limited [20,21] and their effectiveness tends to diminish over time [22]. Therefore, it is imperative to acquire a thorough comprehension of the motivational factors and decision-making mechanisms that underpin behaviors related to preventing domestic injuries. In the current study, we tested the propositions of the integrated model of self-determination theory (SDT) and the theory of planned behavior (TPB) in predicting parental behavioral adherence to early childhood domestic injury prevention in different cultural contexts: Australia (AU), the United States (US), Singapore (SG), and Hong Kong (HK).

### 1.2. Integrated Model of Self-Determination Theory and Theory of Planned Behavior

The integrated model is a theoretical fusion of two prominent social and behavioral theories: self-determination theory (SDT) and the theory of planned behavior (TPB). This integration is substantiated by a wealth of research showing a robust link between the core constructs of the two theories: autonomous motivation from SDT and social cognitive beliefs from the TPB [23]. 

Acting as a more distal self-regulatory process, SDT outlines the motivational origins of the behavior and distinguishes between autonomous and controlled motivation [24]. Autonomous motivation arises when individuals engage in behaviors that align with their personal values, interests, and sense of self. They experience a sense of freedom and personal agency in their choices, viewing their actions as genuine expressions of their authentic selves. On the other hand, controlled motivation emerges when individuals are driven by externally-referenced reasons, for example, external pressure, introjections, or obligations. Their behaviors are contingent upon external control, and once those controlling factors are removed, their commitment to the behavior tends to diminish. Autonomous motivation is considered a more adaptive and sustainable form of motivation because autonomously motivated individuals are more likely to persist with the behavior and obtain more positive outcomes such as improved performance and enhanced psychological well-being [25]. According to SDT, social climate plays a crucial role in determining the quality of motivations an individual experiences. Environments that fulfill and support the basic psychological needs of autonomy, relatedness, and competence are more likely to foster autonomous motivation and promote behavioral persistence.

The TPB provides a more immediate explanation of behavior by highlighting the central role of intention as a mediator between an individual’s social–cognitive belief system and their actual engagement in behavior [26]. The theory suggests that intention is shaped by personal-, social-, and control-related beliefs regarding the behavior. Attitude reflects an individual’s evaluation or appraisal of the behaviors. Subjective norms pertain to the perceived social influences or expectations exerted by significant others to enact a behavior and the individual’s motivation to meet their expectations. PBC represents the individual’s assessment of personal capacity, resources, and the presence of barriers that may facilitate or hinder the execution of the behavior. Individuals are more likely to develop intention and engage in certain behaviors when they perceive their actions as positive, socially acceptable, and feasible.

### 1.3. Integrated Model and Injury Prevention

The tenets of the integrated model have been tested in a wide range of health behaviors, for instance, physical activities [27], binge drinking [28], and healthy eating [29]. The model has also been used to explain behaviors regarding injury management [30]. The intervention study of Lee, Standage, Hagger and Chan [30] demonstrated that the perceived psychological need for support from physical education (PE) teachers promoted autonomous motivation in sports injury prevention. This in turn led to enhanced attitudes, subjective norms, and PBC and, subsequently, heightened intention and adherence toward sport injury-preventive behaviors in high school students.

Aside from its application in the sports injury prevention context, the integrated model has been shown to be effective in predicting behavioral adherence toward prevention against other types of injuries including sunburn [31] and occupational injuries [32,33]. Other unintentional injury prevention-related studies focusing on SDT or the TPB also provide evidence for the motivational and decision-making processes outlined by the integrated model respectively. In support of SDT, it was found that employees’ intrinsic motivation was positively associated with their safety compliance and behaviors in various workplace settings [34,35]. Similarly, autonomous motivation in athletes was consistently found to be predictive of their intention and engagement in sports injury prevention [34,36]. For the TPB, it was shown that pool owners’ intentions and adherence to supervising their young children around pool areas and restricting their pool access were predictable based on their attitude, subjective norms, and PBC toward drowning preventive behaviors [37,38]. Moreover, TPB-based education to mothers appeared to be effective in improving their adoption of childhood domestic injury prevention for their toddlers [39]. In the context of childhood domestic injury prevention, interventions that focus solely on educating caregivers about the importance of protective measures or providing them with the knowledge and tools to perform safety behaviors are not always effective on their own [11]. The integrated model of SDT and TPB is particularly relevant in this regard as it explicitly elucidates how caregivers’ motivational orientations and their behavioral beliefs can directly and indirectly shape their domestic injury prevention behaviors.

### 1.4. Cross-Cultural Examination of the Integrated Model

Previous cross-cultural studies have examined the concepts of the integrated behavior–change model, and their findings from various cultures generally support the robustness of the relationships between the psychological factors from SDT and the TPB in various health settings such as physical activity [40] and education [41]. The original conceptualization of SDT posits that since basic psychological needs were selected for their adaptive survival values through evolution, they are considered as innate and universal [42]. Similarly, for the TPB, Ajzen (1985) proposed that the relationships between the three socio-cognitive constructs and intention would remain consistent across cultures. However, as research has established that culture, albeit distally, infiltrates and impacts the more proximal contexts and, in turn, influences norms, preferences, values, and judgments [43], these claims of universality have been questioned [44]. In particular, researchers have been investigating how collectivistic and individualistic societies may lead to cross-cultural variations in SDT and the TPB. Individuals from cultures with individualistic societies (e.g., Western Europe) typically perceive themselves as independent, autonomous, and distinct from others or social groups. In contrast, individuals from collectivistic societies (e.g., Asia and Eastern Europe) tend to embrace a sociocentric identity and view themselves as more socially sensitive and interdependent with their social groups [45].

With regards to SDT, an early cross-cultural study found that the level of autonomy support received by both Bulgarian employees from state-owned companies (with a collectivistic background) and US employees (with an individualistic background) could predict task motivation and well-being [46]. These findings have been replicated in other cross-cultural studies, including those focused on physical activity [40] and academic achievement [47], thus supporting the generalizability of the theory. Furthermore, a recent review conducted by Ryan, et al. [48], who summarised 60 SDT meta-analyses spanning the topics of work, health, and education, revealed that culture did not significantly moderate the positive relations between basic psychological needs and self-determined forms of motivation. In terms of the TPB, the existing body of research generally supports the positive effect of socio-cognitive beliefs on intention in both collectivist and individualistic cultures, however, the specific contribution of these beliefs to intentions appears to be different. Studies on physical activity [49,50], gambling [51], bone marrow donation [52], and entrepreneurial career intentions [53] have shown that individuals from collectivistic cultures are more influenced by subjective norms and less influenced by personal attitudes than individuals from individualistic cultures. While some studies have examined the cultural effects on SDT and TPB independently, there is a dearth of research exploring the cultural differences within the integrated model. Examining the integrated model in a cross-cultural context could help determine its generalisability and reveal potential differences in the weighting of constructs for the intention of domestic injury prevention.

### 1.5. Present Study

The aim of the current study is to examine the applicability of the integrated model of SDT and TPB in explaining caregivers’ injury prevention behaviors for safeguarding their 2–6-year-old children from domestic injuries. We focused on early childhood because statistics have shown that children in this age group are more vulnerable to domestic injury than older children or adolescents [7,54]. Our research focused on four societies with diverse cultural norms, namely AU, HK, SG, and the US. These societies were chosen for two main reasons: (1) they represent cultures that traditionally emphasize individualism (AU and US) and collectivism (HK and SG), and (2) unintentional injuries are a significant cause of childhood mortality and hospitalizations in these societies [55,56,57,58].

Firstly, to ensure the validity of our measurements and the comparability of results across different societies, we tested the measurement invariance of the standardized scales used in the integrated model. This allowed us to investigate the generalisability of the measures and verify that the survey items were interpreted similarly by participants from various societies and that our measurements accurately reflected the underlying theoretical constructs.

Secondly, drawing from the integrated model framework, we formulated and examined the following four hypotheses across societies (H1–H4).

**H1.** 
*Parents’ perception of psychological need support provided by their family members would be significantly and positively associated with their autonomous motivation in childhood domestic injury prevention.*


**H2.** 
*Parents’ autonomous motivation in childhood domestic injury prevention would be significantly and positively related to the social–cognitive constructs from TPB.*


**H3.** 
*Parents’ social–cognitive beliefs (subjective norms: H3^SN^; perceived behavioral control: H3^PBC^; attitude: H3^ATT^) would form a significant positive association with their intention to prevent childhood domestic injury. In other words, parents with higher subjective norms, PBC, and better attitudes would form stronger intention to carry out childhood domestic injury prevention.*


**H4.** 
*Parents’ intention would be directly and positively associated with their behavioral adherence to prevent childhood domestic injury.*


Thirdly, the relationships among the constructs of the integrated model were compared between societies to investigate whether these patterns of effect are consistent across cultures. Specifically, based on prior cross-cultural research findings [46,47,49,50,51,52,59], we had the following hypotheses regarding the potential similarities and differences among societies (H5).

**H5a.** 
*Compared to individualistic societies (i.e., AU and US), subjective norms would be a stronger predictor of intentions for caregivers in collectivist societies (i.e., HK and SG).*


**H5b.** 
*Compared to collectivist societies (i.e., HK and SG), personal evaluation such as attitude would be a stronger predictor of intentions for caregivers in individualistic societies (i.e., AU and US).*


**H5c.** 
*The strengths of other relationships in the integrated behavior–change model (i.e., psychological need support → autonomous motivation, autonomous motivation → perceived behavioral control, and intention → behavioral adherence) would be invariant across the societies.*


## 2. Materials and Methods

### 2.1. Study Design and Setting

This was a cross-sectional, observational study conducted to examine the relationships among constructs of the integrated model of SDT and TPB regarding childhood domestic injury prevention behaviors. To recruit participants, the researchers contracted with Qualtrics, an online survey platform, to reach respondents across four societies between December 2022 and January 2023.

### 2.2. Participants 

A total of 2059 caregivers (mean age = 35.97 years old, SD = 7.03, male = 49.88%) were recruited from 4 countries (N = AU: 500; HK: 552; SG: 507; US: 500). Criteria for eligible participant included: (1) had at least one child aged between 2 to 6 years; (2) were literate in Chinese or English for questionnaire completion; and (3) the said child was not diagnosed with any medical conditions and physical disabilities. Informed consent was obtained from all subjects involved in the study, and the participants received compensation through Qualtrics after survey completion. The study was ethically approved by the Human Research Ethics Committee at the Education University of Hong Kong (ref: 2022-2023-0033; date of approval: 9 November 2022).

### 2.3. Sampling Methods and Sample Size

The study utilized a crowdsourced participant sampling method, recruiting participants through the existing database of the online survey platform Qualtrics. The expected sample size was 474 participants per society, determined by a priori power analysis. This was estimated by power analysis based on an effect size of 0.19 (taken from the lower boundary of the effect size from our prior study of grandparental adherence towards childhood domestic injury prevention in Hong Kong (subjective norms to intention) [60], alpha of 0.05, power of 0.80 in a structural model with 7 latent variables and 30 indicators) [61]. 

### 2.4. Variables

The 6-item short form of the Health Care Climate Questionnaire (HCCQ) was employed to measure the extent to which caregivers perceived their family members supported their psychological needs. The scale was originally developed to measure the perceived psychological need support from significant others [62] and has been used in various injury-prevention contexts [32,33,63]. The questionnaire was revised for the childhood domestic injury prevention context by starting with the following stem: “To prevent my child(ren) from domestic injuries, …”, and is followed by the item, for example, “I feel that other family members have provided me choices and options.” Participants responded on a 7-point Likert scale (1 = strongly disagree and 7 = strongly agree).

The autonomous motivation subscale was taken from the Treatment Self-Regulation Questionnaire [TSQR, [64], 6 items] for the evaluation of autonomous motivation in childhood domestic injury prevention. The scale has been used to study motivation behind healthy behavior engagement, including injury prevention contexts, and has demonstrated satisfactory reliability and validity [65]. The survey was adopted to make reference to childhood domestic injury prevention, which results in questions with the following stem: “I want to prevent my child(ren) from domestic injuries because …”. An example item was: “I feel that I want to take responsibility for my child(ren)’s health”. Participants responded on a 7-point Likert scale (1 = not at all true and 7 = very true).

Social–cognitive constructs including subjective norms (3 items), perceived behavioral control (6 items), attitude (6 items), and intention (3 items) were examined using the TPB questionnaire [66]. The Chinese version of the scale was developed in previous injury prevention studies, which provided support for its score reliability and validity [33,63,67]. The items were adapted to the context of the current study with the following stem: “Reducing the likelihood of childhood domestic injury in the forthcoming month is something that …”. Example items were “Most people who are important to me think that I should…” (for subjective norms), “I could do if I want to.” (for PBC), and “I intend to…” (for intention). Participants responded on a 7-point Likert scale (1 = strongly disagree and 7 = strongly agree). Items in the attitude subscale were phrased with the stem “Reducing the likelihood of childhood domestic injury in the forthcoming month is something that is…” and participants indicated different levels of affect (e.g., 1 = worthless and 7 = valuable).

The shortened version of the Self-Reported Injury Prevention Adherence Scale was used to evaluate the participants’ frequency (1 item) and effort (1 item) in preventing domestic injuries in their children. The items were “How often do you take measures against domestic injury for your child(ren)?” and “How much effort do you put on preventing domestic injury for your child(ren)?”, respectively. This scale was initially developed to assess individuals’ adherence to sports injury prevention [33] and was adopted for other injury settings (e.g., occupational injury prevention [34]). Participants responded on a 7-point Likert scale (1 = never/minimum effort and 7 = very often/maximum effort).

All questionnaires were adapted for the childhood domestic injury prevention context and presented in Chinese or English (chosen by the participants). The full scales used in the study are shown in Appendix A. Apart from the main constructs in the SDT and TPB models, demographic information of the parents (e.g., age, gender, educational background, marital status, employment status, household members, and household income) and the child (e.g., age, gender, and school grades), a brief history of domestic injuries (e.g., injury causes and types), and caregiving details (e.g., total hours of childcare provided by the respondent and other caregivers) were also collected. In particular, income data were transformed into 10 income levels (10–100th percentiles) according to the four societies’ income census data [68,69,70,71].

### 2.5. Statistical Methods

The measurement validity of the integrated model comprised psychological need support, autonomous motivation, socio-cognition factors, intention, and adherence and was evaluated through single-group confirmatory factor analysis (CFA). Multigroup CFA (MG-CFA) was used to test measurement invariance across four societies [72]. In brief, the configural model (no parameter constraints), metric model (constrained factor loadings), and scalar model (constrained factor loadings and intercepts) were tested and compared [73]. Structural equation modeling (SEM) and multigroup SEM (MG-SEM) with a robust maximum likelihood estimation method were employed to examine the model fit and parameter estimates of the hypothesized pathways for each society and across four societies, respectively (Mplus version 8.1). Chi-square difference tests and Wald tests were used to compare the path coefficients of the societies in the MG-SEM.

Missing values were minimized by setting forced responses on the online survey, resulting in less than 0.1% total missing data. Missing data analysis of all of the variables used in the model showed that the pattern of missing data was completely random (Little’s MCAR test: χ^2^ [195] = 159.92, *p* = 0.97). Various goodness-of-fit indices, such as the Root Mean Square Error of Approximation (RMSEA), the Comparative Fit Index (CFI), the Tucker–Lewis Index (TLI), and the Standardized Root Mean Square Residual (SRMR), were employed to test the overall suitability of the models. To evaluate the adequacy of the model fit, standard threshold values were employed: CFI and TLI values exceeding 0.90 and RMSEA and SRMR values below 0.08 [74]. When assessing measurement invariance through model fit comparison indices, ΔCFI ≤ 0.020, ΔRMSEA and ΔSRMR ≤ 0.030 between the configural and metric models were used to determine metric invariance, while ΔCFI ≤ 0.010 and ΔRMSEA and ΔSRMR ≤ 0.015 between metric and scalar models were used to evaluate scalar invariance following metric invariance [75,76].

## 3. Results

### 3.1. Participants’ Characteristics

Most of the respondents were parents (98.45%), with the remaining participants acting as guardians for their children (e.g., grandparents and family relatives). A majority of our participants were married (81.61%) and had attained an educational level of university bachelor’s degree or above (56.39%). As most of them were full-time employees (father: 84.85%; mother: 55.51%), the amount of time they provided childcare varied (mean = 60.97 h per week; SD = 55.28 h per week). More than half of our participants’ children (mean age = 4.63; SD = 1.33 months; male = 49.88%) had started school (54.98%).

Approximately 27.44% of the participants reported that their child had encountered domestic accidents in the past 4 months, and 63.72% of these accidents led to injuries. Across all four societies, the most common external causes of injuries were falls (49.20%) and mechanical force (17.17%), which resulted in light injuries such as soft tissue injuries (28.32%) or minor head injuries (17.52%).

### 3.2. Preliminary Analysis

The internal consistency of all measures was found to be satisfactory within and across four societies, with Cronbach’s alpha ranging from 0.81 to 0.93. Significant correlations were observed between certain demographic and caregiving variables and the main variables in the integrated model. These variables included participants’ gender, marital status, employment status, household income, children’s age, gender, school enrollment, number of children in the household, and total hours of childcare provided by the respondent and all caregivers per week. To account for their potential confounding effects on the model pathways, these variables were included in the modelw as covariates. The descriptive statistics, score reliability coefficients, and zero-order correlations of these variables are presented in Table 1.

### 3.3. Measurement Invariance across Societies

All of the MG-CFA models of the integrated model (configural, metric, and scalar) demonstrated satisfactory overall model fit, as indicated by the fit indices in Table 2. All loadings of the items on their corresponding latent factors are highly significant (*p* < 0.001) in all societies. To establish the metric invariance of the scales, the configural and metric models of the integrated models were compared. The results showed no significant differences between these models (∆RMSEA = 0.002; ∆CFI = −0.003), fully supporting the metric invariance of the integrated model across societies. To establish the scalar invariance of the scales, the scalar model of the integrated models was compared to the metric model. The comparison did not show a significant difference between models (∆RMSEA = −0.002, ∆CFI = 0.001), providing support for scalar invariance. Strong measurement invariance of the integrated model was supported, indicating that not only are the factor loadings invariant but that the item intercepts of the constructs are also equivalent across the studied societies. This allows meaningful comparisons to be made regarding the structural paths of the latent constructs across different groups.

### 3.4. Structural Pathways of the Integrated Model (H1 to H4)

The results of the MG-SEM configural model showed that the proposed integrated model had an acceptable fit to the data (χ^2^ = 3426.01 [df = 2594], CFI = 0.95, TLI = 0.95, RMSEA = 0.03 [90% CI = 0.02 to 0.03], and SRMR = 0.051). Path estimates largely supported our hypotheses (see Figure 1). As hypothesized, positive and significant effects of psychological need support on autonomous motivation were found in all societies (H1, β = 0.65–0.84, *p* < 0.01). Autonomous motivation in turn showed a positive and significant association with subjective norms (H2^SN^, β = 0.60–0.78, *p* < 0.01), perceived behavioral control (H2^PBC^, β = 0.75–0.92, *p* < 0.01), and attitude (H2^ATT^, β = 0.82–0.93, *p* < 0.01). Consistent with our proposed paths, these three social–cognitive beliefs were found to be positively and significantly related to intention in three societies—AU, SG, and US (H3^SN^, β = 0.36–0.68, *p* < 0.01; H3^PBC^, β = 0.23–0.42, *p* = 0.04; H3^ATT^, β = 0.14–0.26, *p* = 0.04). However, contrary to our hypothesis, caregiver’s attitude in Hong Kong did not significantly relate to their intention toward domestic injury prevention (H3^ATT^, β = 0.09, *p* = 0.08). Finally, in support of H4, there was a positive and significant relationship between intention and behavioral adherence to childhood domestic injury prevention (β = 0.73–0.87, *p* < 0.01). Figure 1 presents the standardized path coefficients for the configural model of the integrated model.

### 3.5. Invariance of Structural Path Coefficients across Societies (H5)

To investigate whether the relationships of the main variables in the integrated model hold true across different societies, we first restricted the direct effects of each path to be invariant across societies and then compared the restricted and unrestricted models with chi-square difference tests. Consistent with our hypothesis (H5a), our analysis revealed significant differences between the restricted and unrestricted models of the pathway from subjective norms to intention (χ^2^ = 14.61 [df = 3], *p* < 0.01). To further probe this distinction, Wald tests were conducted to compare the effects of these pathways across societies. Notably, when compared to all other societies, subjective norms were found to be a weaker predictor of intention in the US. On the other hand, the association between subjective norms and intention was significantly stronger in HK compared to AU. Contrary to our expectations, we did not find support for H5b. Despite the non-significant effect of attitude on intention in Hong Kong, no differences were observed when comparing the models that either restricted or unrestricted this path (χ^2^ = 5.59 [df = 3], *p* = 0.13). Unexpectedly, contrary to our hypothesis (H5c), we found significant differences in two other pathways: psychological need support → autonomous motivation (χ^2^ = 22.30 [df = 2], *p* < 0.01), autonomous motivation → perceived behavior control (χ^2^ = 10.48 [df = 3], *p* = 0.01). Wald tests revealed that the effect of psychological need support on autonomous motivation was significantly stronger in Asian societies (i.e., HK and SG) compared to Western societies (i.e., AU and US). In addition, a heightened effect of autonomous motivation on perceived behavior control was observed in SG when compared with Western societies, while the difference between HK and Western societies was not statistically significant. The differences in other path coefficients across societies were not statistically significant. Table 3 and Table 4 respectively present the model fit statistics of the restricted/unrestricted models and the results of the Wald tests comparing the path coefficients between societies.

## 4. Discussion

The present study provides original evidence for the application of the integrated model of SDT and the TPB in explaining childhood domestic injury prevention across the four studied societies (AU, HK, SG, and the US). Firstly, our study demonstrates that the measures used for assessing the constructs within the integrated model are invariant across societies. Additionally, our results revealed that the proposed relationships between psychological need support, autonomous motivation, socio-cognitive beliefs, intention, and adherence to childhood domestic injury prevention were consistently positive and statistically significant in most societies, thereby largely supporting the motivational antecedents and decision-making process outlined in the model. Notably, we also observed some variations in the extent to which the components of SDT and the TPB contributed to the model across different societies, suggesting potential cultural influences on the theoretical framework. Taken together, these findings lent support to the overall cross-generalisability of the integrated model, while also highlighting some slight variations in its predictive power in different societies.

As a prerequisite of model comparison and interpretation across groups [77], our findings on full scalar invariance suggest that caregivers in different societies ascribed equivalent meanings to the instruments of the integrated model in the context of childhood domestic injury prevention. This finding is congruent with prior cross-cultural studies that independently tested the measurement models of SDT and the TPB in the context of physical activities [49,50], academic performance [47], and entrepreneurial intention [53]. The results also affirmed that the measurement yielded consistent measures of the same attribute in diverse cultures, substantiating the universality of the integrated model [78]. Overall, the establishment of strong measurement invariance allowed valid cross-societal comparisons in the later analyses.

In agreement with our hypotheses (H1–H4), the primary finding from the current study is the rather consistent patterns of relationships among the SDT and TPB variables and their predictive effects on the intention and adherence to childhood domestic injury preventive behaviors across the four societies studied. These findings align with the tenets of the integrated model of SDT and TPB [23] and are comparable to the findings in sports and occupational injury prevention contexts [33,63]. In all societies, caregivers who receive more psychological need support from other family members endorse a higher level of autonomous motivation in childhood domestic injury prevention. This finding provides additional empirical support to the notion that psychological needs are innate and universal, and their satisfaction leads to self-determination [79]. Additionally, a moderate to substantial portion of the variance (45–58%) in the socio-cognitive beliefs was accounted for by autonomous motivation across societies. This suggests that caregivers who engage in injury prevention for self-determined reasons might have a predisposition to develop beliefs that align with these motives. Moreover, the caregivers’ normative, personal, and control beliefs regarding domestic injury prevention were positively related to their intention. The results echoed findings that subjective norms from friends and relatives [80], attitude and beliefs about the outcomes, and perceived barriers and preventability of injury [11,81] were major determinants of parental injury prevention behavior. Overall, the results largely support the cross-cultural application of the integrated model in the explanation of caregivers’ adoption of early childhood domestic injury prevention.

One caveat is that we observed an exception to the principles of TPB in HK, where the effect of attitude on intention did not reach statistical significance. This contradicts previous TPB research on sports and occupational injury prevention [33,63] and most parent-to-child health behaviors [82], where attitude is a significant determinant of the formation of intentions. Our results indicate that Hong Kong caregivers’ personal evaluation of childhood domestic injury-preventive behaviors may not link to their intention to implement such preventive measures. Unlike injury prevention-related TPB studies conducted in other regions (e.g., Brazil [83] and AU [37]), among the three socio-cognitive beliefs, attitude seems to have the smallest effect among TPB-based injury prevention models [30,63] in studies conducted in HK. The weaker link between attitude and intention in HK may be explained by the greater emphasis on collectivist values in this culture. In more collectivist societies like Hong Kong, the influence of social norms and expectations (i.e., subjective norms) may play a more central role in shaping intentions compared to personal evaluations (i.e., attitudes) [45]. Consistent with this, our results indicate that whereas attitude was insignificant in predicting intention, the association between subjective norms and intention in HK was the strongest among all of the studied societies. Future studies may investigate the cross-cultural factors that result in the variation in the contribution of TPB constructs to behavioral intention.

Aside from the non-significant effect of attitude on intention in Hong Kong, the few other cross-cultural differences identified were predominantly centered on the strength of the relationships among the constructs in the integrated model. In line with our hypothesis (H5a), we observed a stronger association between subjective norms and intention in cultures with a collectivistic orientation than in cultures with an individualistic orientation. This is unsurprising given that individuals in collectivist societies tend to prioritize group values over personal values and are more inclined to adjust their personal beliefs to align with societal norms and expectations [84]. Consequently, subjective norms exert a greater explanatory power on intentions in cultures with collectivistic orientation. Similar patterns have emerged from cross-cultural studies examining other health-related behaviors, such as physical activity [49], healthy eating [50], and smoking [85]. Our study corroborates the findings of previous research, providing further evidence for the importance of subjective norms in intention formation within collectivist cultural groups, particularly in the context of childhood domestic injury prevention.

Lastly, contrary to our hypothesis (H5c) that the strengths of effect would remain consistent across societies in other paths, we found that the relationship between basic psychological needs and autonomous motivation was stronger in societies that endorsed collectivistic orientation (i.e., HK and SG) than in societies that endorsed individualistic orientation (i.e., AU and US). These results were also inconsistent with the review findings of Ryan et al. [48], who suggested that the effects of autonomy support were not moderated by cultural differences. The discrepancy between our findings and that of the meta-analyses may be attributed to two potential reasons. First, among the three meta-analyses included in the review, Yu, et al. [86] did not directly examine the specific relationship between basic psychological needs and autonomous motivation, but instead, they emphasized the association between autonomy and subjective well-being. The nuanced differences in the constructs being investigated may have influenced the cultural significance of the results. Secondly, while our study focused exclusively on the context of domestic injury prevention, the other two meta-analyses in the review examined motivation in broader life domains [86] and work settings [87], respectively, which may have led to different conclusions. Particularly, in the case of domestic injury prevention, one notable difference between Western and Asian societies is the degree of involvement of family support in childcare. Compared to Western countries, southeast Asian households rely more heavily on family members for the provision of childcare, with a higher percentage of grandparents taking care of their grandchildren for longer periods of time [88]. Greater involvement of family members in daily caregiving may result in an increased need for psychological need support. Indeed, some qualitative studies have found that both parents and grandparents desire autonomy during childcare in Hong Kong [89] and Singapore [90]. They might believe that higher autonomy could potentially prevent and resolve conflicts when contrasting beliefs and values arise. Therefore, psychological need support might be imperative to the caregivers’ autonomous motivation, particularly in Asian societies.

Our study employed a rigorous statistical method to establish cultural invariance in the integrated model of SDT and the TPB. However, a few limitations should be noted. Firstly, due to the cross-sectional and correlational nature of the study, it is not possible to draw causal inferences regarding the relationships of the constructs. Secondly, although the measures used in this study have been widely validated, our reliance on self-reported measures may introduce biases such as response bias and reporting bias stemming from social desirability [91]. Thirdly, our study used societal membership as a proxy of culture. Although this practice is common among cross-cultural studies, it should be cautioned that cross-cultural differences do not always equate to cross-national differences, as studies have demonstrated that other factors, such as profession, socio-economic status, and globalization, also play a role in shaping cultural values [92]. Future studies can utilize certain instruments (e.g., Horizontal & Vertical Individualism & Collectivism scale) to assess individuals’ cultural orientation [93]. Moreover, it is noteworthy that there might exist more cultural dimensions than the spectrum of collectivism–individualism, for example, power distance and uncertainty avoidance, which may have affected the constructs in the integrated model differently [94]. Fourthly, in order to collect data in countries beyond our primary study locations, we leveraged the use of an online survey platform and its participant database to reach eligible respondents. Although this method of data collection is commonly employed in studies with wide geographical scope [95], large sample size, or/and narrowly defined participant criteria [96,97,98] (in this case, caregivers of 2–6-year-old children), this crowdsourcing approach inherently introduces the potential loss of data quality [99]. Lastly, while our study focused on the preventative behaviors in domestic injuries, it might be informative for future research to broaden the scope and explore the motivational and decision-making processes underlying the prevention of other prevalent unintentional injuries, such as road injuries or drowning, in order to design targeted interventions applicable to a wider range of injury prevention efforts.

## 5. Conclusions

Our study preliminarily examines the integrated model of SDT and the TPB in the context of early childhood domestic injury prevention. We provide initial evidence about the robustness of the model across four different societies. Specifically, our results demonstrate that the positive relationships among psychological need support, autonomous motivation, socio-cognitive beliefs, intention, and behavior adherence to childhood domestic injury prevention were primarily consistent in both Western (i.e., AU and US) and Asian (i.e., HK and SG) societies. The identified differences found between Western and Asian societies may also provide valuable insights for designing culturally tailored and effective interventions. Based on the distinctive predictive pattern of attitude, subjective norms, and perceived behavioral control between the societies, we speculate that interventions aimed at promoting injury prevention among parents in Asian and Western societies could be different. In Asia, interventions could benefit from the focus on social exchange, facilitating discussions, and providing information about the social approval of injury prevention measures; that from Western societies may emphasize the recognition of personal attitude and the development of a sense of control in the prevention of domestic injury among young children. Similar to previous integrated model-based intervention research [30], future research could adopt an experimental intervention design to examine the model’s efficacy in driving sustained behavior change in the context of childhood domestic injury prevention. Additionally, exploring the applicability of the integrated model in other cultural contexts, beyond the Western and Asian societies in the current study, can further enhance our understanding of its generalizability.

## Figures and Tables

**Figure 1 behavsci-14-00701-f001:**
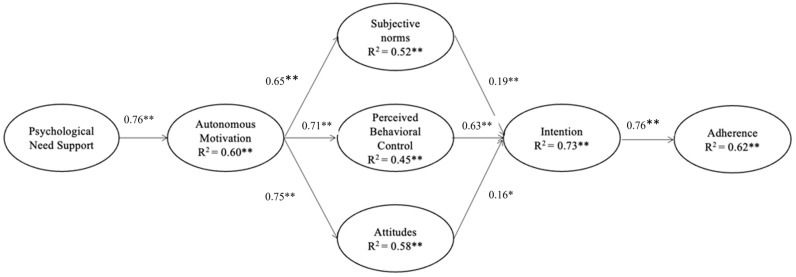
Path estimates for the integrated model of self-determination theory and theory of planned behavior in childhood domestic injury prevention for four societies. Note: For clarity, the paths associated with the control variables (i.e., participants’ gender, marital status, employment status, household income, children’s age, gender, school enrollment, number of children in the household, and total hours of childcare provided by the respondent and all caregivers per week) are omitted. The path coefficients shown in the figure are arranged in the following order: Australia, Hong Kong, Singapore, and the United States. * *p* < 0.05, two-tailed. ** *p* < 0.001, two-tailed.

**Table 1 behavsci-14-00701-t001:** Descriptive statistics, zero-order correlations, and reliability coefficients of the variables (N = 2059).

	1	2	3	4	5	6	7	8	9	10	11	12	13	14	15	16	17
1. Needs support	--																
2. Motivation	0.65 **	--															
3. SN	0.54 **	0.60 **	--														
4. PBC	0.55 **	0.66 **	0.61 **	--													
5. Attitude	0.59 **	0.80 **	0.58 **	0.64 **	--												
6. Intention	0.54 **	0.70 **	0.77 **	0.66 **	0.69 **	--											
7. Adherence	0.45 **	0.60 **	0.58 **	0.51 **	0.58 **	0.72 **	--										
Parental Variables																	
8. Gender	0.05 *	0.12 **	−0.04	0.05 *	0.09 **	0.09 **	0.10 **	--									
9. Marital status	0	0.08 **	-0.04	0.06 **	0.11 **	0.10 **	0.07 **	0.15 **	--								
10. Father—employment	0	0.05 *	0	−0.04	0.08 **	0.05 *	0.03	0.13 **	0.41 **	--							
11. Mother—employment	0.02	0.08 **	−0.01	0.08 **	0.08 **	0.06 **	0.06 **	0.03	0.07 **	0.04 *	--						
Household Variables																	
12. No. of children	0	0.05 *	−0.03	0.04	0.07 **	0.02	0.02	0.14 **	0.10 **	0.11 **	0.16 **	--					
13. Income	0.02	−0.01	−0.01	−0.01	−0.04	−0.04	−0.08 **	−0.03	−0.24 **	−0.22 **	−0.10 **	−0.11 **	--				
14. Total hours per week	0.03	0.13 **	0.06 **	0.06 **	0.13 **	0.10 **	0.08 **	0.13 **	0.04	0.04	0.12 **	0.10 **	0.03	--			
15. Hours per week	0.05 *	0.14 **	0.08 **	0.09 **	0.15 **	0.14 **	0.09 **	0.36 **	0.17 **	0.13 **	0.22 **	0.19 **	−0.01	0.75 **	--		
Children Variables																	
16. Child age	−0.02	−0.05 *	−0.07 **	−0.03	−0.05 *	−0.07 **	−0.08 **	−0.08 **	−0.06 **	0.01	−0.02	0.14 **	0.03	−0.04 *	−0.05 *	--	
17. Child gender	−0.04 *	−0.03	−0.01	−0.02	−0.01	−0.01	0	−0.23 **	−0.04	−0.04	0.01	−0.05 *	0.04	−0.06 **	−0.09 **	0.04	--
18. Child study	−0.03	−0.03	−0.06 *	−0.03	−0.08 **	−0.07 **	−0.05 *	−0.06 **	−0.15 **	−0.06 **	−0.07 **	0	0.02	−0.04	−0.11 **	0.46 **	0.01
Mean	5.69	6.12	5.57	5.7	5.9	5.78	5.51	--	--	--	--	1.77	6.77	109.1	60.97	4.56	--
SD	1.15	1	1.35	1.07	1.09	1.28	1.4	--	--	--	--	0.94	2.99	86.39	55.28	1.32	--
Cronbach’s alpha	0.93	0.94	0.90	0.84	0.91	0.93	0.85	--	--	--	--	--	--	--	--	--	--

Note. * *p* < 0.05, two-tailed. ** *p* < 0.01, two-tailed. Need support = psychological need support. Motivation = autonomous motivation. SN = subjective norms. PBC = perceived behavioral control. Father—employment = employment status of the child’s father. Mother—employment = employment status of the child’s mother. No. of children = number of children within the household. Income = percentile ranking of household income, obtained from official government census data. Total hours per week = total hours of childcare provided by all caregivers per week. Hours per week = total hours of childcare provided by the respondent per week. Child age = the age of the children (in years). Child study = school enrolment status of the child.

**Table 2 behavsci-14-00701-t002:** Multigroup confirmatory factor analysis: Model fit statistics for the integrated model across societies.

MG-CFA	N	χ^2^ (df)	*p*	RMSEA (90% CI)	CFI	TLI	SRMR	∆RMSEA	ΔCFI	∆SRMR
Model 1: Single-group CFA	2059	1642.65 * (391)	<0.001	0.039 [0.037 0.041]	0.955	0.950	0.098	--	--	--
Model 2: Configural Invariance	2059	2260.60 * (1442)	<0.001	0.033 [0.031 0.036]	0.974	0.969	0.055	--	--	--
Model 3: Metric Invariance	2059	2426.28 * (1504)	<0.001	0.035 [0.032 0.037]	0.971	0.966	0.066	0.002	−0.003	0.011
Model 4: Scalar Invariance	2059	2459.74 * (1573)	<0.001	0.033 [0.031 0.036]	0.972	0.969	0.066	−0.002	0.001	0

Note. * *p* < 0.05, two-tailed.

**Table 3 behavsci-14-00701-t003:** Multigroup structural equation modeling: Model fit statistics for the integrated model across societies (with restricted paths).

MG-SEM	Model Goodness-of-Fit							Chi-Square Difference Test	
Path Restricted	N	χ^2^ (df)	*p*	RMSEA (90% CI)	CFI	TLI	SRMR	χ^2^ (df)	*p*
No paths restricted	2059	3426.01 * (2594)	<0.001	0.026 [0.024 0.028]	0.95	0.95	0.051	--	--
Need support → Motivation	2059	3445.58 * (2597)	<0.001	0.026 [0.024 0.029]	0.95	0.94	0.072	22.30 * (3)	<0.001 ***
Motivation → SN	2059	3429.33 * (2597)	<0.001	0.026 [0.024 0.028]	0.95	0.95	0.051	3.58 * (3)	0.31
Motivation → PBC	2059	3432.16 * (2597)	<0.001	0.026 [0.024 0.028]	0.95	0.94	0.052	10.48 * (3)	0.015 *
Motivation → Attitude	2059	3430.01 * (2597)	<0.001	0.026 [0.024 0.028]	0.95	0.95	0.051	5.41 * (3)	0.14
SN → Intention	2059	3442.48 * (2597)	<0.001	0.026 [0.024 0.028]	0.95	0.94	0.051	14.61 * (3)	0.002 **
PBC → Intention	2059	3428.55 * (2597)	<0.001	0.026 [0.024 0.028]	0.95	0.95	0.051	2.70 * (3)	0.44
Attitude → Intention	2059	3430.67 * (2597)	<0.001	0.026 [0.024 0.028]	0.95	0.95	0.051	5.59 * (3)	0.14
Intention → Adherence	2059	3428.50 * (2597)	<0.001	0.026 [0.024 0.028]	0.95	0.95	0.051	1.89 * (3)	0.60

Note. * *p* <0.05, two-tailed. ** *p* <0.01, two-tailed. *** *p* <0.001, two-tailed. Need support = psychological need support. Motivation = autonomous motivation. SN = subjective norms. PBC = perceived behavioral control.

**Table 4 behavsci-14-00701-t004:** Comparison of path coefficients between societies for selected structural paths (Wald tests).

Path	Wald χ^2^ (df)					
	AU vs. HK	AU vs. SG	AU vs. US	HK vs. SG	HK vs. US	SG vs. US
Need support → Motivation	14.14 *** (1)	17.09 *** (1)	0.08 (1)	0.01 (1)	13.52 *** (1)	15.69 *** (1)
Motivation → PBC	1.79 (1)	8.57 ** (1)	0.13 (1)	3.09 (1)	0.94 (1)	6.61 * (1)
SN → Intention	9.81 ** (1)	1.79 (1)	6.96 ** (1)	2.45 (1)	36.47 *** (1)	15.02 *** (1)

Note. * *p* < 0.05, two-tailed. ** *p* < 0.01, two-tailed. *** *p* < 0.001, two-tailed. Need support = psychological need support. Motivation = autonomous motivation. SN = subjective norms. PBC = perceived behavioral control.

## Data Availability

The original data presented in this study are openly available in OSF at https://osf.io/te7vs/ (accessible on 11 August 2024).

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
