# Peer review of "Understanding Parental Adherence to Early Childhood Domestic Injury Prevention: A Cross-Cultural Test of the Integrated Behavior–Change Model"

_behavsci, 2024, doi:10.3390/bs14080701_

Round 1

Reviewer 1 Report

Comments and Suggestions for Authors

This study explores the mechanism to prevent domestic injury in small children through the lens of cultural differences/similarities. The study provides a thorough analysis of how the integration of two important health psychology theories ( self-determination theory and theory of planned behavior) can be applied when exploring cross-cultural differences in domestic injury prevention. The sample size of the study was adequate to statistically explore group differences. The authors emphasise the importance of first exploring the validity of the measures and constructs across the studies. Several other hypotheses were tested using appropriate statistical tests. The comprehensive discussion is a key strength of the paper. I have only minor points to be addressed, which I hope will improve the specificity of the reporting in some parts.

Introduction: There is a need to better describe what constitutes “ domestic injury prevention”

Line 48. “However, concerning statistics reveal that approximately one-third of caregivers failed to implement fall prevention measures or adopt general domestic safety precautions for young children”. Please provide a brief description or examples of ‘fall prevention” or “domestic safety precautions” to better contextualise the issue. Also, one-third of caregivers of which population? 

Line 94. Should be TPB rather than PPB

Methods

L222 – Not clear what means ‘compensated by Qualtrics’ 

L229 – “the expected sample of 474 participants per group”, which group the authors are referring to?  Males/females, countries?

L230 – provide information on what is the outcome measure for the present study. Estimate effect size for which outcome? 

Author Response

-----------------------------------------------Reviewer 1-----------------------------------------------

Reviewer 1 Comment 1: Introduction: There is a need to better describe what constitutes “ domestic injury prevention”

Response to Reviewer 1 Comment 1: Thank you for the comment, we have added some examples to better illustrate

            (Page 2, Line 45-50)

Domestic injury prevention includes active supervision, environmental modification, and safety education for the children [11]. Active supervision encompasses of three dimensions (i.e., attention, proximity and continuity) [11]. Caregivers should constantly attend to and are within close proximity of their child. Environmental modification includes limiting the access to or removing hazards, such as keeping medicine cabinet out of reach [11]. Safety education refers to establishing clear behavioural rules for children to follow, like not touching knives [11].

Reviewer 1 Comment 2: Line 48. “However, concerning statistics reveal that approximately one-third of caregivers failed to implement fall prevention measures or adopt general domestic safety precautions for young children”. Please provide a brief description or examples of ‘fall prevention” or “domestic safety precautions” to better contextualise the issue. Also, one-third of caregivers of which population?

Response to Reviewer 1 Comment 2: We have updated the information based on the previous research, coincidentally, both research (conducted in Canada and Hong Kong respectively), showed approximately one-third of caregivers fail to implement the safety precaution.

            (Page 2, Line 55-57)

However, concerning statistics reveal that approximately one-third of caregivers failed to implement fall prevention measures (such as preventing their children from climbing, the use of belt/ highchair and safety gates) or adopt general domestic safety precautions for young children (such as the use of corner protector, safety gates, and door knobs cover) in Canada and HK respectively [14,15].

Reviewer 1 Comment 3: Line 94. Should be TPB rather than PPB

Response to Reviewer 1 Comment 3: Thank you for spotting this. The abbreviation has been changed.

Reviewer 1 Comment 4:  L222 – Not clear what means ‘compensated by Qualtrics’

Response to Reviewer 1 Comment 4: This line was to convey that the participants were compensated indirectly through Qualtrics, as per contracted, a portion of the fee was paid to the company for recruitment and a portion of the fee was paid to the participants as participation fee.

(Page 5, Line 242-243)

Informed consent was obtained from all subjects involved in the study, and the participants received a compensation through Qualtrics after their survey completion

Reviewer 1 Comment 5: L229 – “the expected sample of 474 participants per group”, which group the authors are referring to?  Males/females, countries?

Response to Reviewer 1 Comment 5: Thanks for pointing this out. The group here refers to 474 per region (AU/HK/SG/US) as we are comparing across societies. The line was changed to:

            (Page 5, Line 250)

The expected sample size was 474 participants per society, determined by a priori power analysis.

Reviewer 1 Comment 6: L230 – provide information on what is the outcome measure for the present study. Estimate effect size for which outcome?

Response to Reviewer 1 Comment 6: The outcome measure for the present study is the adherence of domestic injury prevention, the effect size specified for the sample size estimation is the effect size of subjective norms on intention, which is has the lowest effect size among all the paths from our prior study. The additional information is added.

            (Page 5, Line 253)

This was estimated by power analysis based on an effect size of .19 (taken from the lower boundary of the effect size from our prior study of grandparental adherence towards childhood domestic injury prevention in Hong Kong [subjective norms to intention] [67], alpha of .05, power of .80 in a structural model with 7 latent variables and 30 indicators) [68].

Reviewer 2 Report

Comments and Suggestions for Authors

This manuscript on childhood home injury prevention practices is very interesting. It is well- organised and explores important aspects of how parents and carers can prevent injuries at home using a combined approach from two theories, SDT and TPB. Some parts of the study need more detail and clarification to make it easier to understand. Other reviewers may provide comments on the analysis section that I have not addressed here.

Page 3, integrated model - The explanation for the integrated model of SDT and TPB is clear, but it would be helpful to briefly mention why this integration is particularly relevant for studying childhood injury prevention. A sentence or two linking the theory to your study's specific context could strengthen this section.

Page 4, - Hypothesis 3 (H3) needs clarification. Provide a brief explanation or practical example to support the term "significant positive association". Consider providing an example or more context.

Page 9, Line 376-377 - It is interesting that carers' attitudes in Hong Kong did not significantly correlate with their intentions towards domestic injury prevention. In the discussion section, it would be beneficial to discuss possible reasons for this finding in greater detail. What cultural or contextual factors might explain this deviation from the hypothesis?

Page 13, Conclusions - The conclusion indicates that the study "preliminarily examined" the model, implying the need for more robust future studies. It would be helpful to outline specific future research directions or methodological improvements that could strengthen the evidence for the integrated model.

Comments on the Quality of English Language

proofread needed

Author Response

Dear Editor and Reviewers,

We are grateful for the thoughtful feedback and recommendations provided by the editor and reviewers regarding our manuscript titled  “Understanding Parental Adherence to Early Childhood Domestic Injury Prevention: A Cross-Cultural Test of the Integrated Behaviour-Change Model”. The constructive input we received has been invaluable and instrumental in revising and improving our work. We have thoroughly reviewed and incorporated these valuable suggestions into our revised manuscript. The specific corrections and revisions we have implemented are detailed below.

Reviewer 2 Comment 1:
  Page 3, integrated model - The explanation for the integrated model of SDT and TPB is clear, but it would be helpful to briefly mention why this integration is particularly relevant for studying childhood injury prevention. A sentence or two linking the theory to your study's specific context could strengthen this section.

Response to Reviewer 2 Comment 1: Thank you for the advice. We believed the integrated model can strengthen our understanding of childhood domestic injury prevention as current models and interventions that solely offer knowledge, awareness, and tools for safety behaviours may not always translate into actual preventive actions by caregivers. This warrants the need for a deeper understanding of the underlying psychological factors that drive behaviour in this context.

(Page 3, Line 134-140)

In the context of childhood domestic injury prevention, interventions that focus solely on educating caregivers about the importance of protective measures or providing them with the knowledge and tools to perform safety behaviours are not always effective on their own [11]. The integrated model of SDT and TPB is particularly relevant in this regard as it explicitly elucidates how caregivers' motivational orientations and their behavioural beliefs can directly and indirectly shape their domestic injury prevention behaviours.

Reviewer 2 Comment 2: Page 4, - Hypothesis 3 (H3) needs clarification. Provide a brief explanation or practical example to support the term "significant positive association". Consider providing an example or more context.

Response to Reviewer 2 Comment 2:  We have added the following to be more specific:

(Page 4, Line 208-210)

(H3) Parents’ social-cognitive beliefs (subjective norms: H3SN; perceived behavioural control: H3PBC; attitude: H3ATT) would form a significant positive association with their intention to prevent childhood domestic injury. In other words, parents with higher subjective norms, PBC, and better attitude would form stronger intention for carrying out childhood domestic injury prevention.

Reviewer 2 Comment 3: Page 9, Line 376-377 - It is interesting that carers' attitudes in Hong Kong did not significantly correlate with their intentions towards domestic injury prevention. In the discussion section, it would be beneficial to discuss possible reasons for this finding in greater detail. What cultural or contextual factors might explain this deviation from the hypothesis?

Response to Reviewer 2 Comment 3: We appreciate the comment. The lack of a significant association between caregivers' attitudes and intentions in the HK does indeed deviate from the typical pattern seen in TPB studies, where attitude is often a key determinant of behavioural intentions. One plausible explanation for this finding may be related to cultural differences in the relative importance placed on personal evaluations versus social influences in shaping intentions. HK is generally considered a more collectivist society, where the weight of social expectations and perceived pressures from important others may play a more central role in guiding behaviours compared to individualistic settings.

(Page 12, Line 506-512)

Unlike injury prevention-related TPB studies conducted in other regions (e.g., Brazil [91], AU [43] ),  among the three socio-cognitive beliefs, attitude seems to have the smallest effect among TPB-based injury prevention models [33,34] in studies conducted in HK. The weaker link between attitude and intention in HK may be explained by the greater emphasis on collectivist values in this culture. In more collectivist societies like Hong Kong, the influence of social norms and expectations (i.e., subjective norms) may play a more central role in shaping intentions compared to personal evaluations (i.e., attitudes) [51]. Consistent with this, our results indicated that whereas attitude was insignificant in predicting intention, the association between subjective norms and intention in HK was the strongest among all the studied societies.

Reviewer 2 Comment 4: Page 13, Conclusions - The conclusion indicates that the study "preliminarily examined" the model, implying the need for more robust future studies. It would be helpful to outline specific future research directions or methodological improvements that could strengthen the evidence for the integrated model.

Response to Reviewer 2 Comment 4: Thank you for the comment. We agree that additional research is warranted to validate the utility of the integrated model and provide stronger evidence.

(Page 14, Line 601-606)

Similar to previous integrated model-based intervention research [35], future research could adopt an experimental intervention design examine the model’s efficacy in driving sustained behaviour change in the context of childhood domestic injury prevention.  Additionally, exploring the applicability of the integrated model in other cultural con-texts, beyond the Western and Asian societies in the current study, can further enhance our understanding of its generalizability.
